# Traditional and Non-Traditional Clustering Techniques for Identifying Chrononutrition Patterns in University Students

**DOI:** 10.3390/nu18020190

**Published:** 2026-01-06

**Authors:** José Gerardo Mora-Almanza, Alejandra Betancourt-Núñez, Pablo Alejandro Nava-Amante, María Fernanda Bernal-Orozco, Andrés Díaz-López, José Alfredo Martínez, Barbara Vizmanos

**Affiliations:** 1Doctorate in Translational Nutrition Sciences, Department of Food and Nutrition, Centro Universitario de Ciencias de la Salud, Universidad de Guadalajara, Guadalajara 44340, Jalisco, Mexico; gerardo.mora8568@alumnos.udg.mx (J.G.M.-A.); pablo.nava@alumnos.udg.mx (P.A.N.-A.); fernanda.bernal@academicos.udg.mx (M.F.B.-O.); 2Institute of Nutrigenetics and Translational Nutrigenomics, Centro Universitario de Ciencias de la Salud, Universidad de Guadalajara, Guadalajara 44340, Jalisco, Mexico; 3Laboratory of Nutritional Status Evaluation, Department of Food and Nutrition, Centro Universitario de Ciencias de la Salud, Universidad de Guadalajara, Guadalajara 44340, Jalisco, Mexico; 4Center for Educational Research and University Welfare, Department of Philosophical, Methodological and Instrumental Disciplines, Centro Universitario de Ciencias de la Salud, Universidad de Guadalajara, Guadalajara 44340, Jalisco, Mexico; 5Doctorate in Public Health Sciences, Department of Public Health, Centro Universitario de Ciencias de la Salud, Universidad de Guadalajara, Guadalajara 44340, Jalisco, Mexico; 6Unitat de Salut Pública i Epidemiologia Nutricional, Nutrition and Mental Health (NUTRISAM) Research Group, Department of Basic Medical Sciences, Universitat Rovira i Virgili, 43201 Reus, Spain; andres.diaz@urv.cat; 7Centro de Investigación Biomédica en Red Fisiopatología de la Obesidad y la Nutrición (CIBEROBN), Institute of Health Carlos III, 28029 Madrid, Spain; 8Centre for Research in Endocrinology and Clinical Nutrition, Universidad de Valladolid, 47002 Valladolid, Spain; 9Precision Nutrition and Cardiometabolic Health Program, IMDEA-Nutrition Institute (Madrid Institute for Advanced Studies), CEI UAM + CSIC, 28049 Madrid, Spain; jalfredo.martinez@nutricion.imdea.org; 10Department of Human Reproduction, Growth and Human Development, Centro Universitario de Ciencias de la Salud, Universidad de Guadalajara, Guadalajara 44280, Jalisco, Mexico

**Keywords:** chrononutrition, meal timing, young adults, university students, clustering analysis, eating patterns

## Abstract

**Background/Objectives:** Chrononutrition—the temporal organization of food intake relative to circadian rhythms—has emerged as an important factor in cardiometabolic health. While meal timing is typically analyzed as an isolated variable, limited research has examined integrated meal timing patterns, and no study has systematically compared clustering approaches for their identification. This cross-sectional study compared four clustering techniques—traditional (K-means, Hierarchical) and non-traditional (Gaussian Mixture Models (GMM), Spectral)—to identify meal timing patterns from habitual breakfast, lunch, and dinner times. **Methods:** The sample included 388 Mexican university students (72.8% female). Patterns were characterized using sociodemographic, anthropometric, food intake quality, and chronotype data. Clustering method concordance was assessed via Adjusted Rand Index (ARI). **Results:** We identified five patterns (Early, Early–Intermediate, Late–Intermediate, Late, and Late with early breakfast). No differences were observed in BMI, waist circumference, or age among clusters. Chronotype aligned with patterns (morning types overrepresented in early clusters). Food intake quality differed significantly, with more early eaters showing healthy intake than late eaters. Concordance across clustering methods was moderate (mean ARI = 0.376), with the highest agreement between the traditional and non-traditional techniques (Hierarchical–Spectral = 0.485 and K-means-GMM = 0.408). **Conclusions:** These findings suggest that, while traditional and non-traditional clustering techniques did not identify identical patterns, they identified similar core structures, supporting complementary pattern detection across algorithmic families. These results highlight the importance of comparing multiple methods and transparently reporting clustering approaches in chrononutrition research. Future studies should generate meal timing patterns in university students from other contexts and investigate whether these patterns are associated with eating patterns and cardiometabolic outcomes.

## 1. Introduction

Chrononutrition, an emerging field that examines how timing-related aspects of eating impact circadian rhythms and biological processes [1], represents a critical yet understudied area of dietary behavior research. Beyond the established importance of what we eat, growing evidence indicates that when we eat can independently influence cardiometabolic health [2]. This temporal dimension of nutrition encompasses several aspects, including meal timing (i.e., the clock time at which meals are consumed), eating window duration (the interval between the first and last eating occasion), overnight fasting length, and day-to-day variability in these behaviors [3,4].

Most chrononutrition research has focused on individual variables rather than on integrated patterns of temporal eating behavior. Studies in adults analyzing isolated chrononutrition variables have documented associations between late dinner timing and impaired glucose tolerance [5,6], between greater evening energy intake and increased adiposity [7], and between extended eating windows and elevated cardiometabolic risk [8]. In Spanish and Mexican university students, “eating jet lag” (the difference in meal timing between workdays and free days) ≥3.5 h was associated with higher BMI [9].

Only a limited number of studies have moved beyond individual variables to identify integrated chrononutrition patterns using person-centered clustering approaches, assessing their associations with cardiometabolic markers. A latent profile analysis in US adults (*n* = 5228) identified five chrononutrition profiles (typical eating, early finished eating, later heavy eating, extended-window eating, and restricted window eating) with varying associations with sleep health [10]. Using K-means clustering, a study in European children and adolescents (*n* = 2195) identified three temporal patterns (early–often, late–long, and late–infrequent–short), with the late–infrequent–short pattern associated to higher HOMA-IR levels [11]. Similar pattern-based studies have been conducted in Austrian adults [12], Spanish adults [13], Iranian women [14], Norwegian adults with obesity [15], and Taiwanese adults [16]. Notably, all these studies employed a single clustering method without systematically comparing alternative approaches.

Despite the emergence of pattern-based chrononutrition research, key knowledge gaps remain. First, to our knowledge, no study has identified chrononutrition patterns in university students using clustering approaches. University students constitute a population particularly vulnerable to circadian disruption due to irregular schedules, social demands, and lifestyle changes. Second, no studies have identified chrononutrition patterns in Latin American populations, where distinct cultural meal timing norms (e.g., later dinner times, extended midday breaks) may yield patterns different from those observed in European and North American cohorts. Third, there is no methodological comparison of clustering methods for chrononutrition data, which presents unique analytical challenges, including circular time distributions, constrained eating windows, and strong correlations between meal times. Furthermore, this type of analysis would reveal whether all clustering methods generate similar patterns or whether the patterns differ significantly depending on the method used.

The identification of chrononutrition patterns offers distinct advantages over the analysis of individual temporal variables. Pattern-based approaches capture the multidimensional, co-occurring nature of real-world eating behaviors [12], enabling the identification of population subgroups with distinct risk profiles that may be obscured when variables are analyzed independently. Moreover, pattern-based characterization holds greater translational value for public health messaging, as recommendations targeting integrated behavioral profiles may be more actionable targets than those addressing isolated temporal variables.

Chrononutrition patterns can be identified using cluster analysis, an unsupervised machine learning approach that groups observations based on similarity without predefined class labels [17]. Clustering methods can be broadly categorized as traditional approaches, widely available in standard statistical software and computationally straightforward (K-means, Hierarchical clustering with Ward’s linkage), and non-traditional approaches, requiring specialized algorithms and typically implemented in advanced statistical or machine learning libraries (Gaussian Mixture Models, Spectral clustering). With the expansion of computational capacity, non-traditional methods have become increasingly accessible; however, their application in nutritional epidemiology remains limited. Furthermore, there is no evidence to show whether traditional and non-traditional clustering techniques generate similar or significantly different patterns using the same variables and sample size. In dietary pattern research, methodological studies have compared clustering approaches and found that no single algorithm universally outperforms others, with results depending on data structure and sample size [18,19,20]. Agreement between methods is commonly assessed using the Adjusted Rand Index (ARI), with values below 0.8 suggesting substantial differences in cluster assignments [21,22].

The present study addresses these gaps by (1) systematically comparing traditional and non-traditional clustering methods to evaluate whether different analytical approaches yield concordant or discordant pattern structures; (2) providing the first identification of chrononutrition patterns in a young student population; and (3) characterizing these patterns in a Mexican population, contributing the first pattern-based chrononutrition analysis in Latin America.

This methodological comparison is essential to strengthen future chrononutrition research, as the field cannot advance with confidence if varying methods produce inconsistent results without clear understanding of the causes. By providing empirical evidence on the comparative performance of clustering algorithms in chrononutrition research, this study seeks to inform methodological decision-making and advance rigorous chrononutrition pattern analysis in this emerging field.

We hypothesized that traditional and non-traditional methods would perform similarly well in capturing complex temporal eating structures, as reflected by internal validation metrics and concordance in pattern classifications.

Accordingly, this study aimed to compare four clustering techniques, two traditional (K-means, Hierarchical clustering with Ward’s linkage) and two non-traditional (Gaussian Mixture Models, Spectral clustering), for identifying meal timing patterns, a chrononutrition element, in Mexican university students. Secondary objectives included characterizing the identified patterns in terms of chrononutrition variables, sociodemographic and anthropometric indicators, and food intake quality, and evaluating the consistency of these characterizations across clustering methods.

## 2. Materials and Methods

### 2.1. Study Design and Participants

This cross-sectional study is part of a macro-project that aims to analyze food insecurity risk factors and their consequences on lifestyle and cardiovascular health in workers and students at a public university in Mexico [23]. For the present analysis, we focused exclusively on university students.

Participants were recruited using non-probabilistic convenience sampling, selecting readily accessible subjects based on availability and willingness to participate [24]. Study collaborators conducted in-person invitations to students. We attended four classroom groups of each semester in each health study area (nutrition, medicine, psychology, nursing, physical culture and sports, dentistry, and higher university technician programs). Group selection was arbitrary but ensured representation from both the morning shift (*n* = 2 groups) and afternoon shift (*n* = 2 groups). Approximately 3980 to 4975 students were invited across 199 classroom groups (20–25 students each); the response rate was approximately 9.4–11.7% (466 participants). Graduate students in health-related fields and undergraduate students from non-health programs who voluntarily expressed interest in the study were also included. Data collection took place between April 2022 and November 2023.

University students aged 18 years or older were included. Exclusion criteria included pregnancy or lactation, current use of metabolism-altering medications (e.g., corticosteroids, isotretinoin, antiretrovirals, androgens), and active cancer or recent cancer treatment. For this analysis, we included only participants who reported at least one meal timing; seven participants without any meal timing data were excluded, comprising an analytical sample of 459 out of 466 students. Figure 1 presents the participant flow diagram with the inclusion and exclusion criteria.

This research was conducted following the ethical principles of the updated Declaration of Helsinki [25] and the CIOMS International Ethical Guidelines for Health-related Research Involving Humans [26]. The study protocol was approved by the Research, Research Ethics, and Biosafety Committees of the University Center for Health Sciences, Universidad de Guadalajara (approval CI-02322). Given the potential vulnerability of student populations within academic settings (CIOMS Guideline 15) [26], specific protections were implemented: recruitment and data collection were conducted by the research staff independent of teaching activities, participants received explicit assurance that involvement would not influence their academic evaluation, and voluntary participation was emphasized throughout. Written informed consent was obtained from all participants following a detailed explanation of the study objectives, study procedures, potential risks and benefits, and withdrawal rights. To protect participant privacy, all data were coded with unique numerical identifiers.

### 2.2. Data Collection and Variables

#### 2.2.1. Data Collection

Data were collected in two phases, as previously described [23]. First, participants completed online questionnaires covering sociodemographic characteristics, chrononutrition variables, and food intake quality. Prior to survey access, participants viewed an explanatory text emphasizing voluntary participation, assuring confidentiality, and providing contact information for questions. An explicit consent item required affirmative agreement before proceeding. In the second phase, participants attended an in-person appointment for anthropometric measurements conducted by trained personnel using standardized protocols.

#### 2.2.2. Sociodemographic Characteristics

Participants provided information on age, sex, marital status, academic program, semester level, and employment status. Age was analyzed as a continuous quantitative variable (years). Sex was recorded as male or female. Marital status was dichotomized as single (including single, widowed, or divorced) or in a relationship (married or cohabiting). Academic programs were grouped into four categories based on their frequency: nutrition, medicine, psychology, and other programs (nursing, dentistry, physical culture and sports, higher university technician). Semester level was categorized into two groups based on years in university: 1 to 2 years and 3 to 5 years. Employment was analyzed dichotomously (employed vs. unemployed).

#### 2.2.3. Anthropometric Measurements

Trained personnel collected anthropometric data following standardized protocols [27]. Height was measured to the nearest 0.1 cm using a portable stadiometer (Seca^®^ 213, Hamburg, Germany), and weight was measured to the nearest 0.1 kg using a calibrated digital scale (Tanita^®^ BC-568, Arlington Heights, IL, USA). Waist circumference was measured at the midpoint between the lowest rib and the iliac crest using an anthropometric tape (Lufkin^®^, Missouri, TX, USA). Body Mass Index (BMI) was calculated and categorized according to WHO criteria [28].

#### 2.2.4. Food Intake Quality Assessment

Food intake quality was evaluated using the Mini-Survey to Evaluate Food Intake Quality (Mini-ECCA v.2) [29], a 14-item instrument validated in Mexican university students that incorporates photographs as visual aids for portion size estimation. This questionnaire evaluates adherence to Mexican and international dietary recommendations, evaluating water intake, consumption of vegetables, fruits, fish, oilseeds and avocado, legumes, type of meat (red meat, chicken, fish), type of fat (monounsaturated, polyunsaturated, saturated), type of cereal (whole grain, minimally processed, processed and ultra-processed), sweetened beverages, foods not prepared at home, processed foods, sweets and desserts, and alcoholic beverages. Responses follow a Likert scale (never, sometimes, almost always, always, or one of three options for types of meat/fat/cereals). A discriminant analysis classifies participants into three dietary patterns: healthy food intake, habits in need of improvement, or unhealthy food intake.

#### 2.2.5. Chronotype Assessment

Chronotype was assessed using the Morningness-Eveningness Questionnaire (MEQ), a 19-item instrument that evaluates individual circadian preferences through questions about preferred wake and sleep times, peak alertness periods, and self-perceived morningness–eveningness orientation. Total scores range from 16 to 86 points, with participants classified into three chronotypes: morning type (≥59 points), intermediate type (42–58 points), or evening type (≤41 points). The version used was an adaptation and Spanish translation made available through the Center for Environmental Therapeutics [30], which includes modifications to the response scale compared to the original [31].

#### 2.2.6. Chrononutrition Assessment

Participants reported their usual meal timing through open-ended questions asking for the habitual time at which they consume breakfast, lunch and dinner (e.g., “Usual time at which you have breakfast”). This approach captured typical meal timing schedules rather than actual intake on a specific day. Participants who habitually skipped a meal were instructed to indicate “not applicable.”

These open-ended responses underwent a systematic data cleaning process. When participants reported time ranges (e.g., “8:00–9:00 A.M.”), we calculated the midpoint of the range to obtain a single representative value. All meal times were converted to decimal hour format (0–24 h from midnight) for analysis. For the small number of participants who reported eating after midnight, times were converted to values greater than 24 (e.g., 1:00 A.M. was coded as 25.0 h) to maintain the temporal sequence of meals across the 24 h period.

From these primary meal timing data, several chrononutrition variables were derived. Overnight fasting duration was calculated as the time elapsed from dinner to breakfast the following day, computed as [(24 − dinner time) + breakfast time] for standard cases where dinner occurred before midnight, or [breakfast time − (dinner time − 24)] for the small number of cases where dinner was reported after midnight (coded as >24 h). The midpoint of eating was calculated as the temporal center point between breakfast and dinner times [(breakfast time + dinner time)/2], which serves as a validated marker of circadian eating patterns. Inter-meal intervals were calculated as the time differences between consecutive meals: breakfast to lunch (lunch time − breakfast time) and lunch to dinner (dinner time − lunch time).

### 2.3. Statistical Analysis

#### 2.3.1. Descriptive Analyses

All eight continuous chrononutrition variables were assessed for normality using the Shapiro–Wilk test. As all the variables were non-normally distributed (*p* < 0.05), which is consistent with the inherently periodic and circular nature of meal timing data, non-parametric methods were applied. Continuous variables are presented as medians with interquartile range [Q1, Q3], and categorical variables as frequencies and percentages.

For group comparisons, the Mann–Whitney U test or the Kruskal–Wallis test were used as appropiate. When omnibus Kruskal–Wallis tests indicated significant differences, Dunn’s post-hoc test with Bonferroni correction was used for pairwise comparisons. Chi-square tests were applied to categorical variables; when significant, post hoc pairwise comparisons were performed using Fisher’s exact test with Holm correction.

#### 2.3.2. Clustering Methods

To identify distinct meal timing patterns, four clustering algorithms (two traditional distance-based methods and two non-traditional approaches) were applied to the three core meal timing variables (breakfast, lunch, and dinner times). Prior to clustering, variables were standardized (z-scores) to ensure equal weighting in distance calculations, given that the three meal times span different portions of the day. Traditional methods included (1) K-means clustering, which employed Lloyd’s algorithm [32] with Euclidean distance and 50 random initializations to minimize the risk of local optima; and (2) agglomerative Hierarchical clustering using Ward’s minimum variance linkage [33] with squared Euclidean distance. Non-traditional methods included (3) Gaussian Mixture Models (GMM; model-based clustering), fitted using the Expectation-Maximization algorithm [34] with model selection based on the Bayesian Information Criterion (BIC); and (4) Spectral clustering (graph-based) [35], which employed a radial basis function kernel with bandwidth selected by the median heuristic, followed by K-means in the reduced eigenspace. The Principal Component Analysis (PCA) biplot visualization of clustering structure across methods is provided in Appendix A.

Solutions ranging from k = 2 to k = 5 were evaluated for all methods. Optimal cluster number was determined using silhouette coefficient maximization [36] and gap statistic [37] with 500 bootstrap samples. Visual inspection of clustering metrics complemented statistical validation. The final selection of k = 4 was determined by balancing statistical metrics with the clinical interpretability of the identified temporal patterns. Comprehensive validation metrics across methods and k values are provided in Appendix A, with internal validation indices and gap statistic analysis visualized in Appendix A.

To systematically name the identified clusters, we assigned each cluster a P-code (P1–P4) for breakfast, lunch, and dinner based on the relative position of its median meal time compared to all other clusters within each method, where P1 represents the earliest and P4 the latest. When clusters shared the same median meal time, the cluster with the earlier eating midpoint was assigned the lower P-code. Clusters were then named according to the P-code appearing most frequently across the three meals: Early for predominant P1, Early–Intermediate for predominant P2, Late–Intermediate for predominant P3, and Late for predominant P4. One distinct pattern emerged that required a specific descriptor: P2-P4-P4, representing early breakfast but very late lunch and dinner, was termed “Late with early breakfast” to distinguish it from consistently late patterns. This predominance-based nomenclature ensures systematic and interpretable cluster naming across methods. Detailed temporal characteristics and method-specific cluster compositions are provided in Appendix A.

#### 2.3.3. Clustering Performance and Methods Concordance

Cluster quality was assessed using six complementary internal validation indices, following recommendations that multiple indices should be employed as no single validity index provides consistent results across clustering algorithms [38,39]. The silhouette coefficient [36] measures how similar an object is to its own cluster compared to other clusters, with higher values indicating better-defined clusters. The Dunn index [40] quantifies the ratio of minimum inter-cluster distance to maximum intra-cluster distance, with higher values indicating better separation. The Calinski–Harabasz index [41] measures the ratio of between-cluster to within-cluster variance, with higher values indicating better-defined clusters. Hubert’s Gamma statistic [42] measures the correlation between within-cluster and between-cluster distance rankings, with higher values indicating better cluster structure. The Davies–Bouldin index [43] measures the average similarity between each cluster and its most similar cluster, with lower values indicating better cluster compactness and separation. Entropy [44] quantifies the uncertainty in cluster assignments, with lower values indicating more distinct cluster boundaries. Silhouette analysis across all methods is visualized in Appendix A.

To examine the robustness and concordance of identified clusters across methods, we calculated the Adjusted Rand Index (ARI) [45] for all pairwise method comparisons. The ARI measures agreement between two clustering solutions, adjusting for chance, with values of 1 indicating perfect agreement, 0 indicating random agreement, and negative values indicating worse than random agreement. Cross-method cluster assignment patterns are visualized in Appendix A. Bootstrap stability analysis confirmed reproducible cluster assignments across 500 resamples (Appendix A).

Identified clusters were characterized across sociodemographic, anthropometric, derived chrononutrition variables (eating midpoint, eating window, overnight fasting, intermeal intervals), chronotype, and food intake quality using Kruskal–Wallis tests and chi-square tests. For significant omnibus tests, post hoc pairwise comparisons were conducted using Dunn’s test with Bonferroni correction and Fisher’s exact test with Holm correction.

#### 2.3.4. Software and Statistical Significance

All analyses were conducted in R version 4.3.3 [46] using the following key packages: data processing (haven [47], dplyr [48], tidyr [49], hms [50]), statistical testing (stats, FSA [51]), clustering and validation (cluster [52], mclust [53], kernlab [54], factoextra [55], fpc [56], clusterSim [57]), and visualization (ggplot2 [58], patchwork [59]). A random seed (2025) was set for all procedures involving random number generation to ensure reproducibility. The complete analysis code is available at Zenodo (see Data Availability Statement). Additional results, as referenced throughout this section and detailed at the end of the Conclusions, are presented in the Appendix A. Statistical significance was set at *p* < 0.05 (two-tailed) for all analyses.

## 3. Results

### 3.1. Sample Characteristics

The analytical sample consisted of 459 university students with a median age of 20.0 [19.0, 22.0] years, of whom 72.8% were female. Students came from Nutrition (21.4%), Medicine (19.4%), Psychology (13.3%), and other health sciences programs (46.0%), with 59.9% in their first two years of study. Most participants were single (94.1%), and one-third were employed (32.7%). Median BMI was 22.8 [20.7, 26.0] kg/m^2^ and waist circumference was 72.0 [66.5, 79.4] cm. Chronotype distribution (*n* = 454) was predominantly intermediate (70.0%), followed by morning (22.0%) and evening (7.9%) chronotypes (Table 1).

Among these participants, 388 (84.5%) reported consuming all three main meals (breakfast, lunch, and dinner), while 71 (15.5%) reported skipping at least one meal time. Among those with meal time omission, 78.9% skipped only one meal (most commonly breakfast), while 21.1% skipped two meals. Sociodemographic, anthropometric, and chronotype characteristics did not differ significantly between participants with and without meal time omission (all *p* > 0.05), including age, sex distribution, marital status, academic program, semester level, employment status, BMI, waist circumference, and chronotype distribution.

However, food intake quality differed significantly between groups (*p* < 0.001). Participants who skipped one or more meals had a higher prevalence of unhealthy food intake (65.2%) compared to those consuming all three meals (40.0%). Chrononutrition elements also differed significantly between groups. Compared to those who do not skip meals, participants with meal omission exhibited later meal timing for all main meals (breakfast: 09:30 vs. 09:00 h; lunch: 16:00 vs. 15:30 h; dinner: 22:00 vs. 21:30 h), a later eating midpoint (17:00 vs. 15:15 h), shorter eating windows (6.8 vs. 12.5 h), and longer overnight fasting periods (17.2 vs. 11.5 h) (Table 1).

### 3.2. Meal Timing Patterns by Traditional and Non-Traditional Clustering Methods

For the clustering analyses, only the 388 participants who reported all three main meal times were included. All four algorithms identified distinct patterns with significantly different eating schedules (all pairwise comparisons *p* < 0.001), resulting in five categories: Early, Early–Intermediate, Late–Intermediate, Late, and Late with early breakfast (Table 2, Figure 2).

#### 3.2.1. K-Means Clustering

K-means clustering identified four distinct meal timing patterns: Early (*n* = 94, 24.2%), Early-Intermediate (*n* = 68, 17.5%), Late-Intermediate (*n* = 167, 43.0%), and Late (*n* = 59, 15.2%).

The Early pattern was characterized by early intermediate breakfast (09:00 h), with early lunch (14:30 h), and dinner (20:30 h). This pattern represents individuals who maintain earlier eating schedules throughout the entire day.

The Early–Intermediate pattern was notable for exhibiting the earliest breakfast time across all patterns (06:15 h), followed by intermediate timing for lunch (15:30 h) and early intermediate dinner (21:00 h). This pattern characterizes early risers who shift to more conventional meal times as the day progresses.

The Late–Intermediate cluster represented the most prevalent pattern, comprising nearly half of the sample. This cluster was characterized by intermediate and moderately delayed meal times—breakfast at 09:30 h, lunch at 15:30 h, and dinner at 21:30 h—reflecting the most common eating schedule in this university student population.

The Late pattern was defined by substantially delayed timing across all three meals: breakfast at 10:30 h, lunch at 17:00 h, and dinner at 22:30 h. This pattern represents individuals with consistently late eating schedules throughout the entire day.

#### 3.2.2. Hierarchical Clustering

Hierarchical clustering identified four meal timing patterns: Early (*n* = 90, 23.2%), Early–Intermediate (*n* = 96, 24.7%), Late–Intermediate (*n* = 60, 15.5%), and Late (*n* = 142, 36.6%).

The Early pattern was distinguished by the earliest breakfast time observed in these cluster groups (06:45 h), combined with early intermediate timing for lunch (15:00 h) and dinner (20:52 h). This pattern characterizes very early risers who maintain relatively conventional meal times throughout the rest of the day.

The Early–Intermediate pattern exhibited, as its name suggested, an early intermediate breakfast (09:00 h), lunch (15:00 h), and dinner time (21:00 h). This pattern has similar lunch and dinner times to the Early pattern, but the breakfast time is later.

The Late–Intermediate cluster was characterized by early intermediate breakfast (09:00 h) and lunch timing (15:00 h), but a notably late dinner (22:00 h). This pattern is distinguished from Early–Intermediate by its later dinner time (22:00 h vs. 21:00 h), while both share similar daytime meal schedules.

The Late cluster represented the most prevalent pattern in this method, comprising over one-third of the sample. This cluster was defined by delayed meal times—breakfast at 10:00 h, lunch at 16:00 h, and dinner at 22:00 h—representing individuals with consistently late eating schedules.

#### 3.2.3. Gaussian Mixture Models

GMM clustering identified four meal timing patterns with a notably skewed distribution: Early (*n* = 79, 20.4%), Early–Intermediate (*n* = 233, 60.1%), Late with early breakfast (*n* = 17, 4.4%), and Late–Intermediate (*n* = 59, 15.2%).

The Early pattern was characterized by the earliest breakfast timing among these groups (06:30 h) combined with early intermediate lunch (15:00 h) and dinner (21:00 h) times.

The Early–Intermediate cluster dominated this classification, comprising 60.1% of the sample. This pattern was characterized, as its name suggests, by early intermediate breakfast (09:00 h), lunch (15:00 h), and dinner times (21:15 h). Early and Early–Intermediate patterns share similar lunch and dinner times, but the Early–Intermediate pattern has a later breakfast (09:00 h).

The Late with early breakfast pattern represented a unique eating profile not identified by traditional methods. This small cluster exhibited earlier breakfast (07:00 h) but markedly late lunch (17:00 h) and dinner (22:30 h) times, suggesting a dinner-driven late eating pattern where the temporal center of food intake shifts substantially toward evening hours.

The Late–Intermediate group was distinguished by a very late breakfast (11:00 h) followed by lunch (16:30 h) and dinner (22:00 h), both late, but less late than in the Late with early breakfast pattern. Notably, GMM did not identify a pure Late pattern, suggesting the probabilistic nature of this approach may have redistributed participants with consistently late eating into the Late–Intermediate cluster.

#### 3.2.4. Spectral Clustering

Spectral clustering identified four meal timing patterns: Early (*n* = 86, 22.2%), Early–Intermediate (*n* = 125, 32.2%), Late with early breakfast (*n* = 39, 10.1%), and Late–Intermediate (*n* = 138, 35.6%).

The Early pattern has the earliest breakfast (06:43 h), compared with the other clusters, combined with early intermediate lunch (15:00 h) and dinner (21:00 h) times.

The Early–Intermediate cluster was characterized by early intermediate breakfast (09:00 h), lunch (15:00 h), and dinner (21:00 h) times. The Early and Early–Intermediate patterns have similar lunch and dinner times, but the Early pattern differs in breakfast time (06:43 h vs. 09:00 h).

The Late with early breakfast pattern exhibited a distinctive temporal profile similar to that identified by GMM, though with larger representation. This cluster showed early breakfast (08:00 h) but markedly late lunch (17:30 h) and dinner (22:30 h) times, suggesting an eating pattern where food intake becomes progressively more delayed as the day advances.

The Late–Intermediate cluster represented the most prevalent pattern in this method, comprising 35.6% of the sample. This cluster exhibited late breakfast timing (10:00 h), followed by late lunch (16:00 h) and dinner (22:00 h) times, but these were not as late as the Late with early breakfast pattern. Like GMM, Spectral clustering did not identify a pure Late pattern with consistently delayed timing across all three meals.

While cluster proportions varied by method, reflecting differences in algorithmic approaches to partitioning the data, all four clustering approaches consistently identified these core meal timing patterns. The variation in cluster sizes primarily reflected methodological differences in how boundaries between clusters were defined rather than fundamental disagreement about the underlying patterns themselves.

Having established that all methods identified similar core meal timing patterns, we next examined the sociodemographic composition of participants in each pattern.

### 3.3. Sociodemographic Characteristics by Meal Timing Patterns

No statistically significant differences were identified in age or employment status between clusters across any of the four clustering methods (all *p* > 0.05) (Table 3).

In K-means (traditional method), sex distribution showed significant differences (*p* = 0.012), with Early–Intermediate having the lowest proportion of females (57.4%) compared to Late (79.7%) and Late–Intermediate (76.6%) patterns. Academic program distribution approached significance (*p* = 0.050).

In Hierarchical (traditional method), the academic program showed significant heterogeneity (*p* = 0.008); Late–Intermediate having the lowest proportion of other programs (23.3%). Sex distribution was not significant.

In GMM (non-traditional method), sex distribution showed a significant overall difference (*p* = 0.022), though no specific pairwise comparisons remained significant after Holm correction. Academic program showed significant differences (*p* = 0.044). The Late with early breakfast pattern was notably distinct, with a high concentration of Medicine students (52.9%) compared with early patterns. None of the Nutrition students were included in this pattern.

In Spectral (non-traditional method), no significant differences were identified in sex distribution (*p* = 0.059) or academic program (*p* = 0.118).

### 3.4. Chrononutrition Variables

Chrononutrition variables by clustering method are presented in Table 4. Eating midpoint, eating window duration, overnight fasting, and inter-meal intervals (breakfast-to-lunch and lunch-to-dinner) showed statistically significant differences across clusters within each method (all *p* < 0.001).

Participants in the Early pattern demonstrated a consistent eating midpoint before 15:00 (13:45–14:36) across all four methods. In the Hierarchical, GMM, and Spectral methods, this pattern showed extended eating windows (14.0–14.3 h) with short overnight fasting (9.7–10.0 h). In contrast, K-means identified a shorter eating window (11.8 h) with prolonged overnight fasting (12.2 h). Despite these differences, all Early clusters exhibited consistently long breakfast-to-lunch intervals (6.0–8.2 h), with three of four methods showing intervals ≥ 8.0 h.

Participants in the Early–Intermediate pattern showed heterogeneity across methods. K-means Early–Intermediate displayed an extended eating window (14.9 h) with short overnight fasting (9.1 h) and a prolonged breakfast-to-lunch interval (9.0 h). In contrast, Hierarchical, GMM, and Spectral Early–Intermediate exhibited more compressed profiles with moderate eating windows (11.5–12.0 h), overnight fasting of 12.0–12.5 h, and breakfast-to-lunch intervals of 6.0 h.

Participants in the Late–Intermediate pattern exhibited eating midpoints after 15:00 (15:30–16:30) across all four methods, reflecting delayed meal timing. Eating windows ranged from 11.5 to 13.0 h with overnight fasting of 11.0–12.5 h. K-means and Hierarchical Late–Intermediate showed breakfast-to-lunch intervals of 6.0 h.

Participants in the Late pattern, identified only by traditional methods (K-means and Hierarchical), exhibited eating midpoints after 15:00 (15:45–16:15). Eating windows were 12.0–12.2 h with overnight fasting of 11.8–12.0 h. This pattern showed a short lunch-to-dinner interval (5.0–5.5 h), indicating compressed evening meal timing.

The Late with early breakfast pattern, identified only by non-traditional methods (GMM and Spectral), showed wide eating windows (13.0–15.5 h) combined with short overnight fasting periods (8.5–11.0 h). The breakfast-to-lunch interval was notably prolonged (9.0–10.0 h), reflecting a temporal structure where early breakfast is followed by delayed subsequent meals.

Regarding eating midpoint, this variable showed a clear temporal progression across patterns: Early (before 15:00; 13:45–14:36) → Early–Intermediate (around 15:00; 13:45–15:15) → Late–Intermediate (after 15:00; 15:30–16:30) → Late (after 15:00; 15:45–16:15).

Regarding chronotype, distribution differed significantly across clusters in all four methods (all *p* ≤ 0.001). Morning-type individuals were overrepresented in Early patterns (34.2–37.2%) compared to Late and Late–Intermediate patterns (1.7–16.3%). Intermediate chronotypes predominated across all clusters (54.8–86.4%). Evening-type individuals represented only 7.9% of the sample, limiting statistical power for between-cluster comparisons.

### 3.5. Anthropometric and Food Intake Quality by Meal Timing Patterns

Anthropometric measures and food intake quality characteristics by meal timing patterns are presented in Table 5. Two consistent findings emerged across all four clustering methods: (1) no statistically significant differences in BMI or waist circumference (all *p* > 0.05), indicating that anthropometric measures were independent of meal timing in this university student sample; (2) significant differences in overall food intake quality (all *p* ≤ 0.017). In traditional methods, participants in the Early pattern consistently showed the highest proportion of healthy food intake (K-means: 41.5%; Hierarchical: 43.3%), while participants in the Late pattern showed the highest proportion of unhealthy intake (K-means: 50.8%; Hierarchical: 51.4%). In non-traditional methods, participants in the Early pattern also showed the highest healthy intake (GMM: 46.8%; Spectral: 44.2%). However, patterns in unhealthy intake differed: in GMM, no significant post hoc differences were observed, whereas in Spectral, participants in the Late–Intermediate pattern showed the highest unhealthy intake (53.6%). Habits in need of improvement showed no significant post hoc differences across clusters in any method.

### 3.6. Clustering Performance and Method Concordance

Following recommendations for multi-index validation [38], Table 6 presents six complementary internal validation indices for each method, along with the Adjusted Rand Index (ARI) concordance matrix.

Consistent with evidence that no single validity index provides consistent results across clustering algorithms [38], different methods excelled on different metrics: K-means achieved the highest Silhouette coefficient (0.250), Dunn index (0.045), and Calinski–Harabasz index (149.8), indicating better-defined cluster boundaries and between-cluster variance; and GMM achieved the highest Gamma statistic (0.493), lowest (best) Davies–Bouldin index (1.262), and lowest entropy (1.054), indicating optimal balance between cluster compactness, separation, and assignment certainty. Spectral clustering showed competitive but not best performance across metrics. The concordance matrix revealed moderate overall agreement between methods (mean ARI = 0.376). Notably, the highest concordances were observed between traditional and non-traditional method pairs: Hierarchical–Spectral (ARI = 0.485), K-means-GMM (ARI = 0.408), and K-means–Spectral (ARI = 0.402), suggesting complementary pattern detection across algorithmic families. The lowest concordance was between GMM and Hierarchical methods (ARI = 0.271). These results demonstrate that each clustering algorithm emphasizes different aspects of the data structure, supporting the use of multiple validation indices when comparing clustering methods.

## 4. Discussion

To our knowledge, this is the first study to systematically compare traditional and non-traditional clustering methods for identifying chrononutrition patterns. The four methods successfully identified distinct meal timing patterns characterized by differences in breakfast, lunch, and dinner timing. These patterns were named Early, Early–Intermediate, Late–Intermediate, Late, and Late with early breakfast. Cross-method concordance was moderate, with the highest agreement observed between traditional and non-traditional method pairs. Internal validation metrics showed moderate cluster quality, with K-means and GMM achieving the best scores across different metrics. Significant differences were observed across meal timing patterns in chrononutrition variables, food intake quality, chronotype distributions, sex, and academic program. No significant differences were found in BMI, waist circumference, age, or employment status. These findings underscore the importance of methodological transparency when applying clustering techniques to chrononutrition data.

The five meal timing patterns identified (Early, Early–Intermediate, Late–Intermediate, Late, and Late with early breakfast) exhibited distinct temporal signatures. These patterns were identified by both traditional and non-traditional methods, although not identical across them; the only two patterns showing differences were “Late,” which was exclusively identified by traditional methods (K-means and Hierarchical), and “Late with early breakfast,” which was exclusively identified by non-traditional methods (GMM and Spectral). The “Late” pattern was characterized by delayed timing across all three meals, while the “Late with early breakfast” pattern exhibited early breakfast but delayed lunch and dinner, exhibiting the longest eating windows and exceptionally long breakfast-to-lunch intervals, suggesting that model-based and graph-based approaches may uncover subpopulations missed by distance-based methods. Our five-pattern solution differs from Kim et al.’s latent profiles using NHANES data [10], which emphasized breakfast-skipping as a discriminating feature, whereas our analysis in participants with complete meal reports identified variability primarily in eating timing and window duration. This methodological difference may reflect the fact that meal skipping behaviors tend to form distinct clusters that can dominate pattern identification when included in the analysis.

The number of patterns identified in chrononutrition clustering studies depends on the type of variables analyzed. Studies based on energy distribution typically identify two to three patterns [11,12,13,14], as caloric intake tends to concentrate in a limited number of daily periods. Meal timing variables, however, allow for finer temporal distinctions, which may explain why studies using these variables, including ours, tend to identify a greater number of patterns. Consistent with this, our five-pattern solution aligns with Horn et al.’s five meal patterns in Norwegian adults [15] and Chau et al.’s five temporal patterns in Taiwanese adults [16].

Conceptually, the five patterns represent distinct temporal phenotypes with potential relevance for nutritional epidemiology. The Early pattern may reflect individuals aligned with societal schedules and institutional meal availability, while the Late pattern likely captures those with delayed circadian preferences or occupational constraints. The intermediate patterns suggest a *continuum* of meal timing behaviors rather than discrete categories. Notably, the “Late with early breakfast” pattern represents a unique phenotype where morning eating precedes markedly delayed subsequent meals, potentially reflecting compensatory behaviors or irregular schedules. For behavioral research, these patterns offer actionable phenotypes: interventions targeting late eaters could focus on meal timing advancement, while those addressing the “Late with early breakfast” pattern might emphasize meal regularity throughout the day.

Agreement between the four clustering methods was moderate (mean ARI = 0.376). Pairwise comparisons ranged from 0.271 (GMM–Hierarchical) to 0.485 (Hierarchical–Spectral). The highest concordance between Hierarchical and Spectral methods (ARI = 0.485) likely reflects their shared reliance on pairwise similarity matrices. In contrast, the lowest agreement between GMM and Hierarchical methods may reflect that GMM models clusters as probability distributions with flexible shapes, whereas Hierarchical clustering groups observations by distance without distributional assumptions [19]. These concordance levels are comparable to prior dietary pattern comparison studies. For example, Fransen et al. reported moderate agreement (kappa values ranging 0.28–0.56) when comparing three clustering methods (K-means, two-step clustering, and latent class analysis) in European adolescents [60], while Sauvageot et al. found concordance levels of 0.3–0.6 when comparing K-means, K-medians, and Ward’s linkage for dietary patterns [22]. The moderate concordance in our study indicates that while core meal timing patterns are identifiable across methods, cluster enumeration and boundary definitions remain method-dependent.

The moderate internal validation metrics observed across methods are consistent with benchmarks from previous dietary pattern clustering studies [22,60]. K-means achieved the highest Silhouette coefficient, Dunn index, and Calinski–Harabasz index, indicating well-defined cluster boundaries—consistent with Sauvageot et al.’s finding of superior K-means stability for dietary pattern identification [22]. In contrast, GMM achieved the best Davies–Bouldin index, Gamma statistic, and Entropy, suggesting that probabilistic assignments may better capture gradual transitions between meal timing patterns. This pattern of metric-specific winners illustrates that no single validity index provides consistent results across clustering algorithms [38]. The uniformly low Dunn values across all methods reflect substantial within-cluster heterogeneity characteristic of self-reported meal timing data, suggesting that meal timing patterns may represent gradual transitions along a *continuum* rather than discrete categories with sharp boundaries. From a practical perspective, K-means offers computational efficiency and ease of implementation, while Hierarchical clustering provides intuitive dendrograms for visualizing data structure. GMM accommodates clusters of varying shapes and provides probabilistic assignments [19], whereas Spectral clustering excels at identifying non-convex cluster shapes that distance-based methods may miss [35].

Beyond the three meal times used as clustering variables, significant differences emerged in derived chrononutrition variables across patterns. The eating midpoint showed the clearest pattern differentiation, shifting progressively from Early to Late clusters across all methods. Eating window duration and overnight fasting duration also varied across patterns. The consistency of these chrononutrition variable patterns across our four clustering methods suggests that the derived temporal features could capture meaningful biological variation beyond the input meal times.

The alignment between self-reported chronotype and data-driven meal timing patterns provides convergent validity for the clustering solutions. Morning-type individuals were consistently overrepresented in Early and Early–Intermediate clusters compared to Late and Late–Intermediate patterns across all four methods. This is biologically plausible, as chronotype reflects individual differences in circadian phase preference influencing sleep, activity, and eating schedules [31]. Systematic reviews have documented that evening chronotypes tend to have later mealtimes, skip meals more frequently, and distribute greater energy intake toward later times of the day [61,62]. Our findings extend this evidence by demonstrating alignment between data-driven meal timing patterns and self-reported chronotype in young adults.

The most consistent finding across all four clustering methods was the difference in food intake quality between meal timing patterns, with early eaters demonstrating significantly higher healthy food intake compared to late eaters. This difference persisted across all methodological approaches despite variations in cluster sizes and boundaries. While causal inference cannot be drawn from our cross-sectional data, it is plausible that shortened eating windows among late eaters, and reduced food availability during evening hours when institutional cafeterias are closed may differ from daytime options, potentially influencing food choices. Our findings align with Bonaccio et al.’s analysis of 8688 Italian adults from the INHES study, which found that late eaters reported lower adherence to a Mediterranean Diet and higher consumption of ultra-processed foods compared to early eaters [63]. Lesani et al. similarly reported that late eating patterns were associated with lower diet quality scores in Iranian women, with higher morning energy proportion linked to better overall diet quality [14].

We observed no significant differences in BMI or waist circumference among participants across meal timing patterns in any of the four clustering methods. This null finding likely reflects our young sample (median age 20 years) with relatively narrow anthropometric distributions (median BMI 22.8 kg/m^2^), where differences in food intake quality may not yet have translated into anthropometric disparities. Recent meta-analytic evidence similarly reported inconsistent associations between meal timing and anthropometric outcomes in young populations [64]. Horn et al. found no significant anthropometric differences among five meal patterns in adults with obesity, despite daily energy intake differences of up to 2050 kJ [15], suggesting that meal timing–anthropometric relationships may require longitudinal assessment to detect. Studies beginning in young adulthood are needed to determine whether chrononutrition patterns established during this period influence long-term body composition trajectories.

Sociodemographic analyses revealed sex and academic program differences in meal timing patterns, though age and employment status showed no significant differences. Female students were more frequently represented in Late and Late–Intermediate clusters in both the K-means and GMM methods, although the mechanisms underlying remain unclear. Academic program differences were observed in Hierarchical and GMM methods; notably, in GMM, Medicine students were concentrated in the Late with early breakfast pattern (52.9%) while Nutrition students were absent from this. These patterns may reflect program-specific academic demands that could influence meal scheduling.

Key strengths of this study include the systematic comparison of four methodologically diverse clustering approaches using identical input variables and an identical sample, enabling evaluation of method-specific characteristics and concordance. Unlike most dietary pattern studies, which typically rely on a single clustering method, our multi-method approach provided empirical evidence on how algorithmic assumptions affect results. The use of six complementary internal validation metrics (Silhouette coefficient, Dunn index, Davies–Bouldin index, Calinski–Harabasz index, Hubert’s Gamma statistic, and Entropy) alongside external concordance metric (Adjusted Rand Index) ensured robust assessment of clustering quality. Additionally, our focus on a Latin American university student population addresses a geographic gap in chrononutrition research, which has predominantly examined European, North American, or East Asian cohorts [11,12,13]. Requiring complete data on breakfast, lunch, and dinner time ensured that meal timing variability was fully captured. Finally, food intake quality was assessed using the Mini-ECCA v.2, a culturally adapted and validated instrument for Mexican students [29].

Important limitations should be acknowledged. The cross-sectional design precludes causal inference regarding differences in food intake quality across meal timing patterns. Meal timing was self-reported via online questionnaire rather than measured objectively (e.g., ecological momentary assessment or smartphone-based dietary logging), which may introduce recall bias. However, the aim was to determine the usual consumption times and the usual food consumption, rather than the times and consumption of a specific day. Generalizability is limited due to the use of convenience sampling rather than random selection. The exclusion of participants with meal-skipping (*n* = 71, 15.5%) may have removed a meaningful chrononutrition subgroup. Food intake quality was assessed using a brief screening questionnaire (Mini-ECCA v.2) rather than comprehensive dietary assessment methods such as multiple 24 h recalls or food frequency questionnaires, limiting detailed evaluation of specific dietary components. The sample was predominantly female (72.8%), which may limit generalizability to male populations and precluded sex-stratified analyses. These design limitations (cross-sectional assessment, convenience sampling, self-reported information, and a predominantly female university sample) should contextualize the interpretation of observed differences between meal timing patterns and preclude causal claims.

Our findings suggest potential considerations for chrononutrition research methodology. The moderate cross-method concordance observed in this study (mean ARI = 0.376) indicates that different clustering algorithms may identify similar but not identical patterns when applied to meal timing data. Future studies may benefit from comparing multiple methods or explicitly reporting which validation indices were used to select optimal cluster solutions. This study also demonstrates that data-driven clustering based on three meal times (breakfast, lunch, dinner) successfully identifies meaningful meal timing patterns. It is important to note that the temporal midpoint of the eating window used in our clustering differs from the caloric midpoint (the time at which 50% of daily calories have been consumed) described by Teixeira et al. in Brazilian undergraduates [65] and by Baron et al. in their seminal work linking sleep timing to caloric intake [66]. The difference in food intake quality across meal timing patterns suggests that meal timing patterns may have relevance for dietary behavior assessment in young adult populations, though longitudinal studies are needed to establish temporal relationships. Future observational studies should investigate whether meal timing patterns are associated with cardiometabolic risk factors or predict the incidence of cardiometabolic diseases, beyond individual meal timing variables. Cohort studies following university students through career transitions could determine whether meal timing patterns established during young adulthood persist or shift with changing occupational demands. For interventional research, the identified patterns offer specific targets: randomized trials could evaluate whether advancing meal timing in Late-pattern individuals improves cardiometabolic risk factors or diet quality. The multi-method approach demonstrated here could also serve as a template for developing standardized chrononutrition phenotyping protocols applicable across populations.

## 5. Conclusions

This study provides the first systematic comparison of traditional and non-traditional clustering methods for meal timing pattern identification in chrononutrition research. All four methods—K-means, Hierarchical, Gaussian Mixture Models, and Spectral clustering—identified similar meal timing patterns characterized by early-to-late eating progressions, though cross-method concordance was moderate, confirming that algorithm choice influences cluster boundaries while core pattern structures remain consistent. Importantly, the biological validity of these data-driven patterns was supported by their associations with chronotype (morning types in early clusters) and food intake quality (healthier profiles in early eaters), suggesting that clustering captures meaningful circadian-aligned eating behaviors rather than statistical artifacts. For chrononutrition researchers, these findings underscore the importance of reporting which clustering method and validation indices were used, as no single method dominated all metrics. Future studies should explore whether meal timing pattern classification improves prediction of metabolic outcomes beyond individual meal timing variables. Ultimately, this work suggests that data-driven clustering offers a reproducible approach for identifying meal timing phenotypes that may underlie personalized chrononutrition interventions.

## Figures and Tables

**Figure 1 nutrients-18-00190-f001:**
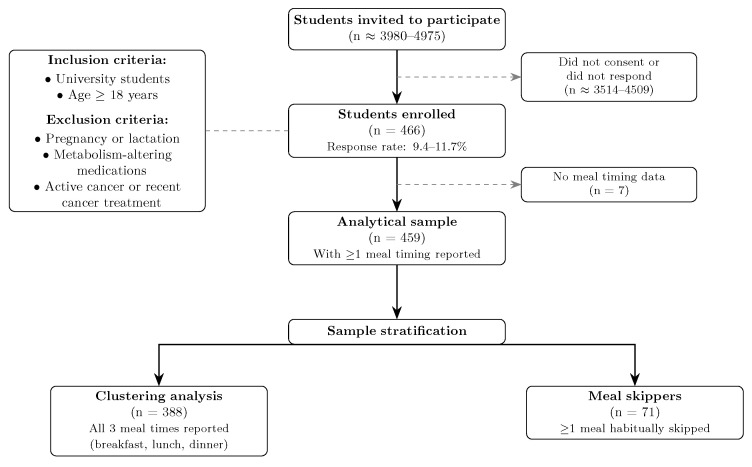
Participant flow diagram (STROBE style). The diagram shows the recruitment process, inclusion and exclusion criteria, and final sample sizes for the analytical sample (*n* = 459) and clustering analysis (*n* = 388). Response rate was approximately 9.4–11.7% (466 participants from 3980–4975 invited).

**Figure 2 nutrients-18-00190-f002:**
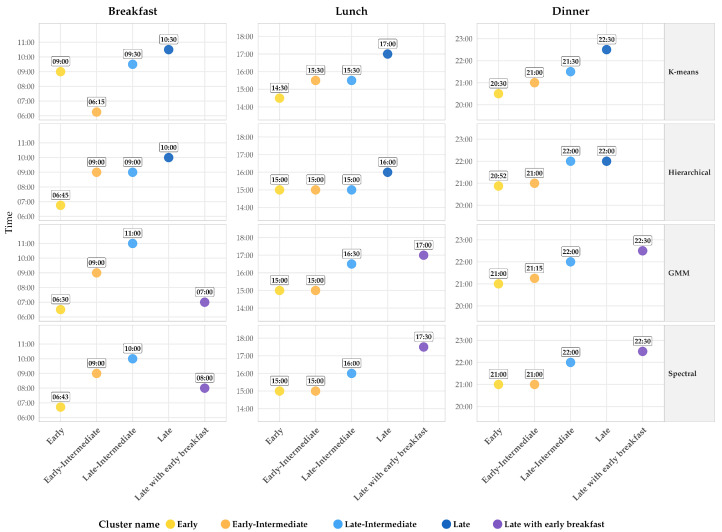
Meal timing patterns by traditional and non-traditional clustering methods (k = 4 clusters). Grid layout: traditional methods (K-means, Hierarchical) and non-traditional methods (GMM, Spectral) × 3 meal times (columns: Breakfast, Lunch, Dinner). Points indicate median times (HH:MM format). Y-axis ranges: breakfast 06:00–11:00 h; lunch 14:00–18:00 h; dinner 20:00–23:00 h. Each color represents a distinct meal timing pattern; dots of the same color across meal columns define the temporal profile of one pattern. For example, in K-means, the Early pattern exhibits breakfast at 09:00 h, lunch at 14:30 h, and dinner at 20:30 h. All methods identified consistent meal timing patterns despite methodological differences in cluster assignments.

**Table 1 nutrients-18-00190-t001:** General characteristics of total participants, and according to whether or not they skip meals.

Characteristic	Total (*n* = 459)	No Meal Skipping (*n* = 388)	≥1 Meal Skipped (*n* = 71)	*p*-Value
*Demographics*				
Age (years)	20.0 [19.0, 22.0]	20.0 [19.0, 22.0]	20.0 [19.0, 21.5]	0.173
Sex *n* (%)				
Female	334 (72.8)	283 (72.9)	51 (71.8)	0.962
Male	125 (27.2)	105 (27.1)	20 (28.2)	
Marital status				
Single	432 (94.1)	364 (93.8)	68 (95.8)	0.711
In relationship	27 (5.9)	24 (6.2)	3 (4.2)	
Academic program				
Medicine	89 (19.4)	78 (20.1)	11 (15.5)	0.319
Nutrition	98 (21.4)	87 (22.4)	11 (15.5)	
Psychology	61 (13.3)	51 (13.1)	10 (14.1)	
Other programs	211 (45.9)	172 (44.3)	39 (54.9)	
Semester level				
1–2 years	272 (59.9)	224 (58.5)	48 (67.6)	0.191
3–5 years	182 (40.1)	159 (41.5)	23 (32.4)	
Employment status				
Employed	150 (32.7)	128 (33.0)	22 (31.0)	0.847
Unemployed	309 (67.3)	260 (67.0)	49 (69.0)	
*Anthropometric measures*				
BMI (kg/m^2^)	22.8 [20.7, 26.0]	22.7 [20.7, 25.8]	23.1 [20.7, 26.4]	0.318
Waist circumference (cm)	72.0 [66.5, 79.4]	71.8 [66.3, 79.0]	72.7 [67.2, 81.1]	0.241
Food intake quality, *n* (%)				
Healthy food intake	126 (27.8)	115 (29.9)	11 (15.9)	<0.001
Habits in need of improvement	129 (28.4)	116 (30.1)	13 (18.8)	
Unhealthy food intake	199 (43.8)	154 (40.0)	45 (65.2)	
Chronotype, *n* (%)	(*n* = 454)	(*n* = 385)	(*n* = 69)	
Morning type	100 (22.0)	90 (23.4)	10 (14.5)	0.261
Intermediate	318 (70.0)	265 (68.8)	53 (76.8)	
Evening type	36 (7.9)	30 (7.8)	6 (8.7)	
*Chrononutrition variables*				
Breakfast time	09:00 [08:00, 10:00] (*n* = 417)	09:00 [07:54, 10:00]	09:30 [09:00, 10:30] (*n* = 29)	0.015 ^†^
Lunch time	15:30 [15:00, 16:00] (*n* = 449)	15:30 [15:00, 16:00]	16:00 [15:00, 17:00] (*n* = 61)	0.025 ^†^
Dinner time	21:30 [21:00, 22:00] (*n* = 430)	21:30 [21:00, 22:00]	22:00 [21:00, 22:30] (*n* = 42)	0.007 ^†^
Eating midpoint	15:15 [14:30, 16:00] (*n* = 459)	15:15 [14:30, 15:45]	17:00 [13:45, 19:00] (*n* = 61)	0.002 ^†^
Eating window (h)	12.0 [11.0, 13.2] (*n* = 459)	12.5 [11.5, 13.5]	6.8 [6.0, 8.0] (*n* = 61)	<0.001 ^†^
Overnight fasting (h)	12.0 [10.8, 13.0] (*n* = 459)	11.5 [10.5, 12.5]	17.2 [16.0, 18.0] (*n* = 61)	<0.001 ^†^
Breakfast-Lunch interval (h)	6.5 [5.5, 7.5] (*n* = 410)	6.5 [5.5, 7.5]	6.5 [6.0, 7.5] (*n* = 22)	0.722 ^†^
Lunch-Dinner interval (h)	6.0 [5.0, 6.5] (*n* = 420)	6.0 [5.0, 6.5]	6.4 [6.0, 7.5] (*n* = 32)	0.004 ^†^

Data presented as median [Q1, Q3] for continuous variables and *n* (%) for categorical variables. Differences for continuous variables were analyzed using Mann–Whitney U test and for categorical variables using Chi-square test. For ≥1 meal-skipped group: derived chrononutrition variables (eating midpoint, eating window, overnight fasting) calculated using first and last available meal times (minimum two meals required); meal intervals calculated using specific meal pairs. “*n* = ” indicates available participants for each variable. ^†^
*p*-value calculated among participants with available data for the respective variable.

**Table 2 nutrients-18-00190-t002:** Meal timing patterns by traditional (K-means, Hierarchical) and non-traditional (GMM, Spectral) clustering methods (*n* = 388).

**K-means**					
**Mealtime**	**C1 Early** ***n*** **= 94 (24.2%)**	**C2 Early-Intermediate** ***n*** **= 68 (17.5%)**	**C3 Late** ***n*** **= 59 (15.2%)**	**C4 Late-Intermediate** ***n*** **= 167 (43.0%)**	***p*-Value**
Breakfast	09:00 [08:00, 09:30] ^*b*^	06:15 [05:50, 07:00] ^*a*^	10:30 [09:00, 11:00] ^*d*^	09:30 [09:00, 10:00] ^*c*^	<0.001
Lunch	14:30 [14:00, 15:00] ^*a*^	15:30 [14:45, 16:00] ^*b*^	17:00 [16:38, 17:30] ^*d*^	15:30 [15:00, 16:00] ^*c*^	<0.001
Dinner	20:30 [20:00, 21:00] ^*a*^	21:00 [20:45, 21:30] ^*b*^	22:30 [22:00, 23:00] ^*d*^	21:30 [21:00, 22:00] ^*c*^	<0.001
**Hierarchical**					
**Mealtime**	**C1 Early** ***n*** **= 90 (23.2%)**	**C2 Late-Intermediate** ***n*** **= 60 (15.5%)**	**C3 Late** ***n*** **= 142 (36.6%)**	**C4 Early-Intermediate** ***n*** **= 96 (24.7%)**	***p*-Value**
Breakfast	06:45 [06:00, 07:26] ^*a*^	09:00 [08:49, 09:30] ^*b*^	10:00 [09:00, 10:30] ^*c*^	09:00 [09:00, 09:30] ^*b*^	<0.001
Lunch	15:00 [14:00, 15:30] ^*a*^	15:00 [14:30, 15:00] ^*a*^	16:00 [16:00, 17:00] ^*c*^	15:00 [15:00, 15:33] ^*b*^	<0.001
Dinner	20:52 [20:00, 21:30] ^*a*^	22:00 [21:30, 22:30] ^*b*^	22:00 [21:30, 22:30] ^*b*^	21:00 [20:30, 21:00] ^*a*^	<0.001
**Gaussian Mixture Models**					
**Mealtime**	**C1 Early** ***n*** **= 79 (20.4%)**	**C2 Late with Early Breakfast** ***n*** **= 17 (4.4%)**	**C3 Late-Intermediate** ***n*** **= 59 (15.2%)**	**C4 Early-Intermediate** ***n*** **= 233 (60.1%)**	***p*-Value**
Breakfast	06:30 [06:00, 07:00] ^*a*^	07:00 [06:15, 07:30] ^*a*^	11:00 [10:30, 11:30] ^*c*^	09:00 [09:00, 10:00] ^*b*^	<0.001
Lunch	15:00 [14:30, 15:30] ^*a*^	17:00 [17:00, 17:30] ^*d*^	16:30 [16:00, 17:15] ^*c*^	15:00 [15:00, 16:00] ^*b*^	<0.001
Dinner	21:00 [20:22, 21:30] ^*a*^	22:30 [22:00, 22:45] ^*c*^	22:00 [21:30, 22:52] ^*c*^	21:15 [21:00, 22:00] ^*b*^	<0.001
**Spectral**					
**Mealtime**	**C1 Early** ***n*** **= 86 (22.2%)**	**C2 Late-Intermediate** ***n*** **= 138 (35.6%)**	**C3 Early-Intermediate** ***n*** **= 125 (32.2%)**	**C4 Late with Early Breakfast** ***n*** **= 39 (10.1%)**	***p*-Value**
Breakfast	06:43 [06:00, 07:26] ^*a*^	10:00 [09:00, 10:30] ^*c*^	09:00 [08:45, 09:30] ^*b*^	08:00 [07:00, 10:30] ^*b*^	<0.001
Lunch	15:00 [14:04, 15:30] ^*a*^	16:00 [15:30, 16:30] ^*b*^	15:00 [14:30, 15:00] ^*a*^	17:30 [17:00, 17:52] ^*c*^	<0.001
Dinner	21:00 [20:15, 21:30] ^*a*^	22:00 [21:00, 22:00] ^*b*^	21:00 [20:30, 21:30] ^*a*^	22:30 [22:00, 23:00] ^*c*^	<0.001

Each statistical method shows four clusters (C1–C4). Time shown in HH:MM format (24 h clock). Data presented as median [Q1, Q3]. The differences in hours between the clusters were analyzed with Kruskal–Wallis tests. Different superscript letters (^*a, b, c, d*^) indicate significant differences between clusters after post hoc comparisons with Bonferroni correction; clusters sharing the same letter do not differ significantly. Letters are assigned in order of median values, with (^*a*^) indicating the earliest meal time within each row. *p* < 0.05 was considered significant.

**Table 3 nutrients-18-00190-t003:** Sociodemographic characteristics by meal timing pattern.

**K-means**					
**Variable**	**C1 Early** ***n*** **= 94 (24.2%)**	**C2 Early–Intermediate** ***n*** **= 68 (17.5%)**	**C3 Late** ***n*** **= 59 (15.2%)**	**C4 Late–Intermediate** ***n*** **= 167 (43.0%)**	***p*-Value**
Age (years)	21.0 [20.0, 23.0]	20.0 [19.0, 22.0]	20.0 [19.0, 22.0]	20.0 [19.0, 22.0]	0.129
Sex, Female	69 (73.4) ^*a*^	39 (57.4) ^*a*^	47 (79.7) ^*b*^	128 (76.6) ^*b*^	0.012
Academic program					
Nutrition	26 (27.7) ^*a*^	15 (22.1) ^*a*^	6 (10.2) ^*a*^	40 (24.0) ^*a*^	0.050
Medicine	9 (9.6) ^*a*^	17 (25.0) ^*b*^	16 (27.1) ^*b*^	36 (21.6) ^*ab*^	
Psychology	13 (13.8) ^*a*^	10 (14.7) ^*a*^	5 (8.5) ^*a*^	23 (13.8) ^*a*^	
Other programs	46 (48.9) ^*a*^	26 (38.2) ^*a*^	32 (54.2) ^*a*^	68 (40.7) ^*a*^	
Employment, Yes	33 (35.1)	17 (25.0)	19 (32.2)	59 (35.3)	0.460
**Hierarchical**					
**Variable**	**C1 Early*****n*** **= 90 (23.2%)**	**C2 Late–Intermediate** ***n*** **= 60 (15.5%)**	**C3 Late** ***n*** **= 142 (36.6%)**	**C4 Early–Intermediate** ***n*** **= 96 (24.7%)**	***p*-Value**
Age (years)	20.0 [19.0, 22.0]	21.0 [19.0, 22.0]	20.0 [19.0, 22.0]	21.0 [19.0, 23.0]	0.794
Sex, Female	59 (65.6)	45 (75.0)	108 (76.1)	71 (74.0)	0.339
Academic program					
Nutrition	19 (21.1) ^*a*^	17 (28.3) ^*a*^	23 (16.2) ^*a*^	28 (29.2) ^*a*^	0.008
Medicine	17 (18.9) ^*a*^	16 (26.7) ^*a*^	33 (23.2) ^*a*^	12 (12.5) ^*a*^	
Psychology	13 (14.4) ^*a*^	13 (21.7) ^*a*^	14 (9.9) ^*a*^	11 (11.5) ^*a*^	
Other programs	41 (45.6) ^*a*^	14 (23.3) ^*b*^	72 (50.7) ^*a*^	45 (46.9) ^*a*^	
Employment, Yes	24 (26.7)	20 (33.3)	50 (35.2)	34 (35.4)	0.531
**Gaussian Mixture Models**					
**Variable**	**C1 Early** ***n*** **= 79 (20.4%)**	**C2 Late with Early Breakfast** ***n*** **= 17 (4.4%)**	**C3 Late–Intermediate** ***n*** **= 59 (15.2%)**	**C4 Early–Intermediate** ***n*** **= 233 (60.1%)**	***p*-Value**
Age (years)	20.0 [19.0, 22.0]	19.0 [19.0, 20.0]	21.0 [19.0, 22.5]	21.0 [19.0, 22.0]	0.197
Sex, Female	52 (65.8) ^*a*^	8 (47.1) ^*a*^	46 (78.0) ^*a*^	177 (76.0) ^*a*^	0.022
Academic program					
Nutrition	18 (22.8) ^*a*^	0 (0.0) ^*a*^	12 (20.3) ^*a*^	57 (24.5) ^*a*^	0.044
Medicine	14 (17.7) ^*a*^	9 (52.9) ^*b*^	12 (20.3) ^*ab*^	43 (18.5) ^*a*^	
Psychology	10 (12.7) ^*a*^	1 (5.9) ^*a*^	5 (8.5) ^*a*^	35 (15.0) ^*a*^	
Other programs	37 (46.8) ^*a*^	7 (41.2) ^*a*^	30 (50.8) ^*a*^	98 (42.1) ^*a*^	
Employment, Yes	23 (29.1)	5 (29.4)	23 (39.0)	77 (33.0)	0.661
**Spectral**					
**Variable**	**C1 Early** ***n*** **= 86 (22.2%)**	**C2 Late-Intermediate** ***n*** **= 138 (35.6%)**	**C3 Early-Intermediate** ***n*** **= 125 (32.2%)**	**C4 Late with Early Breakfast** ***n*** **= 39 (10.1%)**	***p*-Value**
Age (years)	20.0 [19.0, 22.0]	20.0 [19.0, 22.0]	21.0 [19.0, 22.0]	20.0 [19.0, 22.0]	0.401
Sex, Female	56 (65.1)	110 (79.7)	92 (73.6)	25 (64.1)	0.059
Academic program					
Nutrition	18 (20.9)	30 (21.7)	35 (28.0)	4 (10.3)	0.118
Medicine	16 (18.6)	30 (21.7)	19 (15.2)	13 (33.3)	
Psychology	12 (14.0)	16 (11.6)	21 (16.8)	2 (5.1)	
Other programs	40 (46.5)	62 (44.9)	50 (40.0)	20 (51.3)	
Employment, Yes	23 (26.7)	51 (37.0)	41 (32.8)	13 (33.3)	0.475

Each statistical method shows four clusters (C1–C4). Data presented as median [Q1, Q3] for continuous variables and *n* (%) for categorical variables. Differences analyzed with Kruskal–Wallis tests (Bonferroni-corrected post hoc) for continuous variables and Chi-square tests (Fisher’s exact post hoc with Holm correction) for categorical variables. Different superscript letters (^*a, b*^) indicate statistically significant differences between clusters; clusters sharing the same letter do not differ significantly. *p* < 0.05 was considered significant.

**Table 4 nutrients-18-00190-t004:** Chrononutrition variables by derived clustering method.

**K-means**	**C1 Early** ***n* = 94 (24.2%)**	**C2 Early–Intermediate** ***n* = 68 (17.5%)**	**C3 Late** ***n* = 59 (15.2%)**	**C4 Late–Intermediate** ***n* = 167 (43.0%)**	***p*-Value**
Eating Window (h)	11.8 [11.0, 12.5] ^*a*^	14.9 [14.0, 15.5] ^*b*^	12.2 [11.5, 13.0] ^*c*^	12.0 [11.0, 13.0] ^*c*^	<0.001
Overnight Fasting (h)	12.2 [11.5, 13.0] ^*a*^	9.1 [8.5, 10.0] ^*b*^	11.8 [11.0, 12.5] ^*c*^	12.0 [11.0, 13.0] ^*c*^	<0.001
Eating Midpoint	14:36 [14:00, 15:00] ^*a*^	13:45 [13:22, 14:02] ^*b*^	16:15 [15:30, 16:45] ^*c*^	15:30 [15:15, 16:00] ^*d*^	<0.001
Breakfast-Lunch (h)	6.0 [5.0, 6.2] ^*a*^	9.0 [8.0, 9.8] ^*b*^	7.0 [6.0, 8.0] ^*c*^	6.0 [5.5, 6.9] ^*a*^	<0.001
Lunch-Dinner (h)	6.0 [5.0, 6.5] ^*a*^	6.0 [5.0, 6.5] ^*a*^	5.0 [4.8, 6.0] ^*b*^	6.0 [5.5, 7.0] ^*a*^	<0.001
Chronotype, *n* (%)					
Morning	34 (36.6) ^*a*^	25 (37.3) ^*a*^	4 (6.8) ^*b*^	27 (16.3) ^*b*^	<0.001
Intermediate	51 (54.8) ^*a*^	41 (61.2) ^*ab*^	49 (83.1) ^*b*^	124 (74.7) ^*b*^	
Evening	8 (8.6) ^*a*^	1 (1.5) ^*a*^	6 (10.2) ^*a*^	15 (9.0) ^*a*^	
**Hierarchical**	**C1 Early** ***n*** **= 90 (23.2%)**	**C2 Late–Intermediate** ***n*** **= 60 (15.5%)**	**C3 Late** ***n*** **= 142 (36.6%)**	**C4 Early–Intermediate** ***n*** **= 96 (24.7%)**	***p*-Value**
Eating Window (h)	14.0 [13.0, 15.0] ^*a*^	13.0 [12.0, 13.3] ^*b*^	12.0 [11.0, 13.0] ^*c*^	11.5 [11.0, 12.0] ^*d*^	<0.001
Overnight Fasting (h)	10.0 [9.0, 11.0] ^*a*^	11.0 [10.7, 12.0] ^*b*^	12.0 [11.0, 13.0] ^*c*^	12.5 [12.0, 13.0] ^*d*^	<0.001
Eating Midpoint	13:45 [13:30, 14:00] ^*a*^	15:30 [15:15, 16:00] ^*b*^	15:45 [15:30, 16:28] ^*c*^	15:00 [14:45, 15:15] ^*d*^	<0.001
Breakfast–Lunch (h)	8.0 [7.5, 9.0] ^*a*^	6.0 [5.0, 6.0] ^*b*^	6.5 [5.5, 7.5] ^*c*^	6.0 [5.0, 7.0] ^*b*^	<0.001
Lunch–Dinner (h)	6.0 [5.5, 6.5] ^*a*^	7.0 [7.0, 7.8] ^*b*^	5.5 [5.0, 6.0] ^*c*^	5.5 [5.0, 6.0] ^*c*^	<0.001
Chronotype, *n* (%)					
Morning	33 (36.7) ^*a*^	9 (15.0) ^*b*^	17 (12.1) ^*b*^	31 (33.0) ^*a*^	<0.001
Intermediate	54 (60.0) ^*a*^	43 (71.7) ^*ab*^	111 (78.7) ^*b*^	57 (60.6) ^*a*^	
Evening	3 (3.3) ^*a*^	8 (13.3) ^*a*^	13 (9.2) ^*a*^	6 (6.4) ^*a*^	
**GMM**	**C1 Early** ***n*** **= 79 (20.4%)**	**C2 Late with Early Breakfast** ***n*** **= 17 (4.4%)**	**C3 Late-Intermediate** ***n*** **= 59 (15.2%)**	**C4 Early-Intermediate** ***n*** **= 233 (60.1%)**	***p*-Value**
Eating Window (h)	14.3 [13.6, 15.1] ^*a*^	15.5 [14.8, 15.8] ^*b*^	11.5 [10.4, 12.0] ^*c*^	12.0 [11.0, 13.0] ^*d*^	<0.001
Overnight Fasting (h)	9.7 [8.9, 10.4] ^*a*^	8.5 [8.2, 9.2] ^*b*^	12.5 [12.0, 13.6] ^*c*^	12.0 [11.0, 13.0] ^*d*^	<0.001
Eating Midpoint	13:45 [13:21, 14:00] ^*a*^	14:45 [14:22, 15:00] ^*b*^	16:30 [16:04, 17:00] ^*c*^	15:15 [15:00, 15:30] ^*d*^	<0.001
Breakfast-Lunch (h)	8.2 [7.5, 9.0] ^*a*^	10.0 [9.6, 10.8] ^*b*^	6.0 [5.0, 6.5] ^*c*^	6.0 [5.5, 6.8] ^*c*^	<0.001
Lunch-Dinner (h)	6.0 [5.3, 6.5] ^*a*^	5.0 [4.8, 6.0] ^*b*^	5.5 [4.6, 6.0] ^*b*^	6.0 [5.0, 6.8] ^*a*^	<0.001
Chronotype, *n* (%)					
Morning	27 (34.2) ^*a*^	4 (25.0) ^*ab*^	1 (1.7) ^*b*^	58 (25.1) ^*a*^	0.001
Intermediate	49 (62.0) ^*a*^	12 (75.0) ^*ab*^	51 (86.4) ^*b*^	153 (66.2) ^*a*^	
Evening	3 (3.8) ^*a*^	0 (0.0) ^*a*^	7 (11.9) ^*a*^	20 (8.7) ^*a*^	
**Spectral**	**C1 Early** ***n*** **= 86 (22.2%)**	**C2 Late-Intermediate** ***n*** **= 138 (35.6%)**	**C3 Early-Intermediate** ***n*** **= 125 (32.2%)**	**C4 Late with Early Breakfast** ***n*** **= 39 (10.1%)**	***p*-Value**
Eating Window (h)	14.1 [13.5, 15.0] ^*a*^	12.0 [11.0, 12.5] ^*b*^	12.0 [11.0, 13.0] ^*b*^	13.0 [12.1, 15.4] ^*c*^	<0.001
Overnight Fasting (h)	9.9 [9.0, 10.5] ^*a*^	12.0 [11.5, 13.0] ^*b*^	12.0 [11.0, 13.0] ^*b*^	11.0 [8.6, 11.9] ^*c*^	<0.001
Eating Midpoint	13:45 [13:30, 14:00] ^*a*^	15:51 [15:30, 16:15] ^*b*^	15:00 [14:45, 15:25] ^*c*^	15:15 [14:45, 17:00] ^*bc*^	<0.001
Breakfast–Lunch (h)	8.2 [7.5, 9.0] ^*a*^	6.0 [5.5, 7.0] ^*b*^	6.0 [5.0, 6.0] ^*b*^	9.0 [7.0, 10.0] ^*a*^	<0.001
Lunch–Dinner (h)	6.0 [5.5, 6.5] ^*a*^	5.5 [5.0, 6.5] ^*ab*^	6.0 [5.5, 6.8] ^*a*^	5.0 [4.6, 6.0] ^*b*^	<0.001
Chronotype, *n* (%)					
Morning	32 (37.2) ^*a*^	19 (13.8) ^*b*^	35 (28.0) ^*a*^	4 (10.3) ^*b*^	<0.001
Intermediate	52 (60.5) ^*a*^	99 (71.7) ^*ab*^	82 (65.6) ^*ab*^	32 (82.1) ^*b*^	
Evening	2 (2.3) ^*a*^	19 (13.8) ^*b*^	7 (5.6) ^*a*^	2 (5.1) ^*ab*^	

Each statistical method shows four clusters (C1–C4). Data presented as median [Q1, Q3] for continuous variables and n (%) for categorical variables. Chronotype data available for *n* = 385 participants. Differences analyzed with Kruskal–Wallis tests (Bonferroni-corrected post hoc) for continuous variables and Chi-square tests (Fisher’s exact post-hoc with Holm correction) for categorical variables. Different superscript letters (^*a, b, c, d*^) indicate statistically significant differences between clusters; clusters sharing the same letter do not differ significantly. *p* < 0.05 was considered significant.

**Table 5 nutrients-18-00190-t005:** Anthropometric and food intake quality characteristics by meal timing pattern.

**K-means**					
**Variable**	**C1 Early** ***n*** **= 94 (24.2%)**	**C2 Early–Intermediate** ***n*** **= 68 (17.5%)**	**C3 Late** ***n*** **= 59 (15.2%)**	**C4 Late–Intermediate** ***n*** **= 167 (43.0%)**	***p*-Value**
BMI (kg/m^2^)	22.9 [20.8, 26.4]	23.0 [21.1, 25.2]	22.5 [21.0, 27.0]	22.7 [20.1, 25.6]	0.949
Waist circumference (cm)	71.5 [66.4, 78.7]	72.9 [66.5, 79.9]	70.3 [66.7, 80.3]	72.4 [66.2, 78.4]	0.955
Food Intake Quality, *n* (%)					
Healthy food intake	39 (41.5) ^*a*^	31 (45.6) ^*a*^	9 (15.3) ^*b*^	36 (21.6) ^*b*^	<0.001
Habits in need of improvement	28 (29.8) ^*a*^	18 (26.5) ^*a*^	20 (33.9) ^*a*^	50 (29.9) ^*a*^	
Unhealthy food intake	26 (27.7) ^*a*^	18 (26.5) ^*a*^	30 (50.8) ^*b*^	80 (47.9) ^*b*^	
**Hierarchical**					
**Variable**	**C1 Early** ***n*** **= 90 (23.2%)**	**C2 Late-Intermediate** ***n*** **= 60 (15.5%)**	**C3 Late** ***n*** **= 142 (36.6%)**	**C4 Early-Intermediate** ***n*** **= 96 (24.7%)**	***p*-Value**
BMI (kg/m^2^)	22.9 [20.6, 25.6]	22.8 [21.2, 25.5]	22.8 [20.5, 26.5]	22.1 [20.4, 25.6]	0.521
Waist circumference (cm)	72.1 [66.3, 79.2]	72.9 [68.2, 80.0]	72.2 [66.6, 79.2]	71.1 [65.4, 77.2]	0.537
Food Intake Quality, *n* (%)					
Healthy food intake	39 (43.3) ^*a*^	16 (26.7) ^*ab*^	26 (18.3) ^*b*^	34 (35.4) ^*a*^	<0.001
Habits in need of improvement	25 (27.8) ^*a*^	17 (28.3) ^*a*^	42 (29.6) ^*a*^	32 (33.3) ^*a*^	
Unhealthy food intake	26 (28.9) ^*a*^	27 (45.0) ^*ab*^	73 (51.4) ^*b*^	28 (29.2) ^*a*^	
**Gaussian Mixture Models**					
**Variable**	**C1 Early** ***n*** **= 79 (20.4%)**	**C2 Late with Early Breakfast** ***n*** **= 17 (4.4%)**	**C3 Late–Intermediate** ***n*** **= 59 (15.2%)**	**C4 Early–Intermediate** ***n*** **= 233 (60.1%)**	***p*-Value**
BMI (kg/m^2^)	23.0 [20.6, 25.6]	22.8 [21.8, 24.6]	23.0 [20.7, 27.0]	22.5 [20.6, 25.6]	0.605
Waist circumference (cm)	71.8 [66.2, 79.5]	74.8 [69.8, 80.0]	72.0 [67.1, 79.3]	71.6 [66.1, 78.8]	0.396
Food Intake Quality, *n* (%)					
Healthy food intake	37 (46.8) ^*a*^	3 (17.6) ^*ab*^	12 (20.3) ^*b*^	63 (27.0) ^*b*^	0.017
Habits in need of improvement	19 (24.1) ^*a*^	5 (29.4) ^*a*^	21 (35.6) ^*a*^	71 (30.5) ^*a*^	
Unhealthy food intake	23 (29.1) ^*a*^	8 (47.1) ^*a*^	26 (44.1) ^*a*^	97 (41.6) ^*a*^	
**Spectral**					
**Variable**	**C1 Early** ***n*** **= 86 (22.2%)**	**C2 Late–Intermediate** ***n*** **= 138 (35.6%)**	**C3 Early–Intermediate** ***n*** **= 125 (32.2%)**	**C4 Late with Early Breakfast** ***n*** **= 39 (10.1%)**	***p*-Value**
BMI (kg/m^2^)	22.8 [20.4, 25.5]	22.6 [20.0, 26.3]	22.6 [20.8, 25.6]	23.8 [21.4, 26.6]	0.717
Waist circumference (cm)	71.5 [66.3, 78.9]	72.2 [66.0, 80.0]	71.6 [66.3, 78.0]	72.4 [68.4, 81.2]	0.697
Food Intake Quality, *n* (%)					
Healthy food intake	38 (44.2) ^*a*^	26 (18.8) ^*b*^	43 (34.4) ^*a*^	8 (20.5) ^*b*^	<0.001
Habits in need of improvement	23 (26.7) ^*a*^	37 (26.8) ^*a*^	41 (32.8) ^*a*^	15 (38.5) ^*a*^	
Unhealthy food intake	25 (29.1) ^*a*^	74 (53.6) ^*b*^	40 (32.0) ^*a*^	15 (38.5) ^*ab*^	

Each statistical method shows four clusters (C1–C4). Data presented as median [Q1, Q3] for continuous variables and *n* (%) for categorical variables. Differences analyzed with Kruskal–Wallis tests (Bonferroni-corrected post-hoc) for continuous variables and Chi-square tests (Fisher’s exact post hoc with Holm correction) for categorical variables. Different superscript letters (^*a, b*^) indicate statistically significant differences between clusters; clusters sharing the same letter do not differ significantly. *p* < 0.05 was considered significant. BMI = Body Mass Index.

**Table 6 nutrients-18-00190-t006:** Internal validation metrics and concordance between clustering methods (k = 4).

**Method**	**Silhouette ^a^**	**Dunn ^a^**	**Calinski-Harabasz ^a^**	**Hubert’s Gamma ^a^**	**Davies-Bouldin ^b^**	**Entropy ^b^**
K-means	**0.250**	**0.045**	**149.8**	0.477	1.396	1.298
Hierarchical	0.195	0.039	124.5	0.355	1.411	1.341
GMM	0.247	0.038	112.1	**0.493**	**1.262**	**1.054**
Spectral	0.215	0.019	108.1	0.446	1.870	1.297
**Concordance Matrix (Adjusted Rand Index)**						
	K-means	Hierarchical	GMM	Spectral		
K-means	—					
Hierarchical	0.281 *	—				
GMM	0.408 **	0.271 *	—			
Spectral	0.402 **	0.485 ***	0.411 **	—		

^a^ Higher values indicate better clustering quality. ^b^ Lower values indicate better clustering quality. **Bold** indicates the most favorable value among the four methods (highest for ^a^ metrics; lowest for ^b^ metrics). Following Ikotun et al. [38], multiple indices are employed as no single validity index provides consistent results across clustering algorithms. ARI: Method agreement (0 = random, 1 = perfect). Concordance: *** High (>0.45), ** Moderate-High (0.35–0.45), * Low (<0.3). Mean ARI = 0.376.

## Data Availability

The anonymized dataset and reproducible R code (version 1.0.0) are publicly available at Zenodo: https://doi.org/10.5281/zenodo.17795547 (accessed on 2 December 2025).

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
