# Peer review of "Traditional and Non-Traditional Clustering Techniques for Identifying Chrononutrition Patterns in University Students"

_nutrients, 2026, doi:10.3390/nu18020190_

Round 1
Reviewer 1 Report
Comments and Suggestions for Authors
This is an interesting research article with adequate novelty which assesses the chrononutrition patterns of university students. Some points should be addressed.
- Since this study was exclusively performed on university student this should be reported in the title of the manuscript.
- The vast majority of study population concerns female university students. This should be reported in the limitations of the study.
- At the end of the abstract, the authors should add a brief sentences concerning what futures studies could be performed based on the results of the current study.
- In section 2, the authors should add a flow chart diagram, including into it the exclusion and inclusion criteria.
- The final % response rate should be included in the 2nd section.
- The online questionnaire could increase the recall bias. This issue should be reported in the limitations of the study.
- The academic performance and the living status (living alone vs living with their family) could be useful to be incuded in the study. Are there any such data?
- In figure 1, the letters in x-axis should be increased in order to be more visible.
- The discussion section is well-written; however, the authors should try to use a more simple language, which may be more easily understoof to the readers.
Reviewer 2 Report
Comments and Suggestions for Authors
In the manuscript nutrients-4052028, the authors aimed to attract readers' interest by identifying chrononutrition patterns in a young student population, using both traditional and non-traditional clustering methods. The manuscript is based on 65 references from 1948 to 2025, nearly half of which were published in the past 5 years.
Abstract
The abstract is well-organized and briefly presents the essential data. The authors are invited to revise the keywords; they should retain the most relevant ones and include the convenience sampling technique used to select the participants. For example, college students and young adults are doubtful because the authors stated that they "focused exclusively on university students" (lines 105-106). Moreover, clustering analysis is enough, and all details (K-means, hierarchical clustering, Gaussian Mixture Models, and Spectral clustering) can be removed.
Introduction
The literature data are well selected as background for the present study. However, after presenting the literature data, the authors should present the hypotheses, then state the study's aim and the gaps addressed.
Materials and Methods
Generally, the study protocol is clearly presented. All steps are well-illustrated and grouped into sub-subsections with relevant titles.
Line 107. Please check the term "non-probabilistic convenience sampling" (line 107) and include relevant references for this method.
Results
Generally, the presentation of results is accurate and well-organized.
Line 316. Please review the term "Late (early breakfast)" and consider changing it to "Late with early breakfast" for clarity, and update the entire manuscript.
Discussion
Generally, all findings are discussed in detail.
The reviewer suggests beginning this section by presenting the method used for participant recruitment (convenience sampling) and highlighting its advantages.
Conclusions
Please review the last phrase (lines 664-666) and consider changing "demonstrates" to "suggests" and "inform" to "underlie."
Comments on the Quality of English LanguageModerate revision.
Reviewer 3 Report
Comments and Suggestions for Authors
Thank you for the article, however I have some comments, which are intended to be general and constructive:
-
While the analytical approach is comprehensive, briefly summarizing the practical implications of using different clustering techniques in the main text (in addition to supplementary materials) may improve readability for a broader audience.
-
The discussion could further emphasize the conceptual similarities and differences between the identified chrononutrition patterns across methods, highlighting their potential relevance for nutritional epidemiology and behavioral research.
-
Expanding on how these chrononutrition patterns might inform future observational or interventional studies could enhance the applied value of the findings.
-
A concise reinforcement of the study’s limitations—particularly regarding study design and population—may help frame the conclusions more cautiously.
Round 2
Reviewer 1 Report
Comments and Suggestions for Authors
The authors have considerably revised and improved their manuscript.